# Integrating post-validation surveillance of lymphatic filariasis with the WHO STEPwise approach to non-communicable disease risk factor surveillance in Niue, a study protocol

**Adam T. Craig** [1]*, **Harriet Lawford** [1], **Grizelda Mokoia**[2], **Minerva Ikimau**[2], **Peter Fetaui**[2], **Tonia Marqardt**[3], **Colleen L. Lau**[1]

1 Centre for Clinical Research, The University of Queensland, Brisbane, QLD, Australia, 2 Department of Health, Alofi, Niue, 3 National Centre for Immunisation Research and Surveillance, Sydney, NSW, Australia

* adam.craig@uq.edu.au

## Abstract

**Data Availability Statement:** No datasets were generated or analysed during the current study. All

### Background

Lymphatic filariasis (LF), a mosquito-borne parasitic disease caused by three species of filarial worms, was first detected in Niue, a small Pacific Island nation of approximately 1,600 people, in 1954. After extensive efforts involving multiple rounds of Mass Drug Administration, Niue was validated by the World Health Organization (WHO) as having e4liminated LF as a public health problem in 2016. However, no surveillance has been conducted since validation to confirm infection rates have remained below WHO's elimination threshold. WHO has encouraged an integrated approach to disease surveillance and integrating LF post-validation surveillance (PVS) with existing surveys is an anticipated recommendation of the upcoming WHO LF-PVS guidelines. This paper describes a protocol for the implementation of an integrated approach to LF-PVS in Niue as cost-efficient and operationally feasible means of monitoring the disease in the population.

### Methods

The LF-PVS will be implemented as part of a planned national population-based WHO STEPwise approach to non-communicable disease (NCD) risk factor surveillance (STEPS) in Niue. Integration between the LF-PVS and STEPS will occur at multiple points, including during pre-survey community awareness raising and engagement, when obtaining informed consent, during the collection of demographics, risk factor, and location data, and when collecting finger-prick blood samples (for glucose as part of the STEPS survey and LF as part of the LF-PVS). The primary outcome measure for LF transmission will be antigen positivity. Microfilaria slides will be prepared for any antigen-positive cases. Dried blood spots will be prepared for all participants for Multiplex Bead Assays-based analysis to detect anti-filarial antibodies. We estimate a total sample size of 1,062 participants aged 15–69, representing approximately 66% of the population.

relevant data from this study will be made available upon study completion.

**Funding:** This work is funded by the Global Institute for Disease Elimination (FADE2024/03), the (Australian) National Centre for Immunisation Research and Surveillance (NCIRS) and Operational Research and Decision Support for Infectious Diseases Program at the Centre for Clinical Research, the University of Queensland (UQ). Staff from UQ led the study design, the decision to publish and the reparation of the manuscript.

**Competing interests:** The authors have declared that no competing interests exist.

## Conclusions

The results of this study will provide insight into the status of LF in Niue and evaluate the advantages, challenges, and opportunities offered by integrated approaches to disease surveillance.

## Background

Lymphatic filariasis (LF) is a mosquito-borne parasitic disease caused by three species of filarial worms: *Wuchereria bancrofti*, *Brugia malayi* and *Brugia timori* [1]. In the South Pacific, *W. bancrofti* is the parasite responsible for LF [2–4]. The parasites are transmitted to humans following a mosquito bite and subsequently migrate to the lymphatic system, where they mature, mate, and release millions of microfilariae (Mf) into the bloodstream. The early stages of LF infection are often asymptomatic [3], although episodes of acute dermatolymphangioadenitis and acute filarial lymphangitis can occur [5]. If left untreated, secondary bacterial infections and inflammation can occur, causing progressive damage to the lymphatic system. This damage may result in severe manifestations such as lymphoedema, elephantiasis, and scrotal hydrocele [3]. Globally, LF is the second leading cause of chronic disability [6,7].

Transmission of the parasites to humans is facilitated by mosquitoes belonging to the *Anopheles*, *Aedes*, *Culex*, and *Mansonia* genera [6]. In Niue, the primary vector of LF is *Ae. cooki* [8], a diurnal (daytime feeding) mosquito that inhabits a wide range of natural sites including tree holes, leaf axils, coconuts, crab holes and rock holes, as well as artificial containers such as canoes, tyres and drums. *Ae. cooki* is often found at the periphery of villages and tends to rest and feed indoors [4,9].

The Global Programme to Eliminate LF (GPELF), launched by the World Health Organization (WHO) in 2000, is one of the largest public health initiatives worldwide [10,11]. The programme aims to interrupt transmission through mass drug administration (MDA) of anthelmintic medicines and to control morbidity in affected populations [12]. GPELF initially aimed for elimination by 2020 [13]. New milestones and targets have been established in the WHO *Neglected Tropical Diseases Roadmap 2030* [12], which aims for all countries to complete their MDA programmes, implement post-MDA or post-validation surveillance (PVS), and provide a minimum package of care for LF morbidity by 2030 [14]. The goal of MDA is to reduce infection prevalence to a level where transmission is no longer sustainable, with elimination criteria in *Aedes* vector areas set at less than 1.0% antigen prevalence in children aged 6–7 years. In 1999, the regional arm of GPELF, the Pacific Programme to Eliminate Lymphatic Filariasis (PacELF), was initiated in 22 Pacific Island Countries and Territories (PICTs) [3,15].

While eight countries in the Pacific region have achieved validation of LF elimination [16], there is limited evidence to guide the development of effective and efficient PVS strategies [11]. Operational research is needed to determine effective sampling strategies to confirm the presence or absence of LF transmission post-validation, develop cost-effective and timely methods to identify ongoing transmission and ensure long-term sustainability by integrating surveillance into other public health programmes.

### The history of LF surveillance and elimination activities in Niue, 1954 to 2024

Niue is a self-governing island in the South Pacific Ocean, located approximately 660 km southeast of Samoa, 480 km east of Tonga, and 2,400 km northeast of New Zealand (19˚ S 169˚

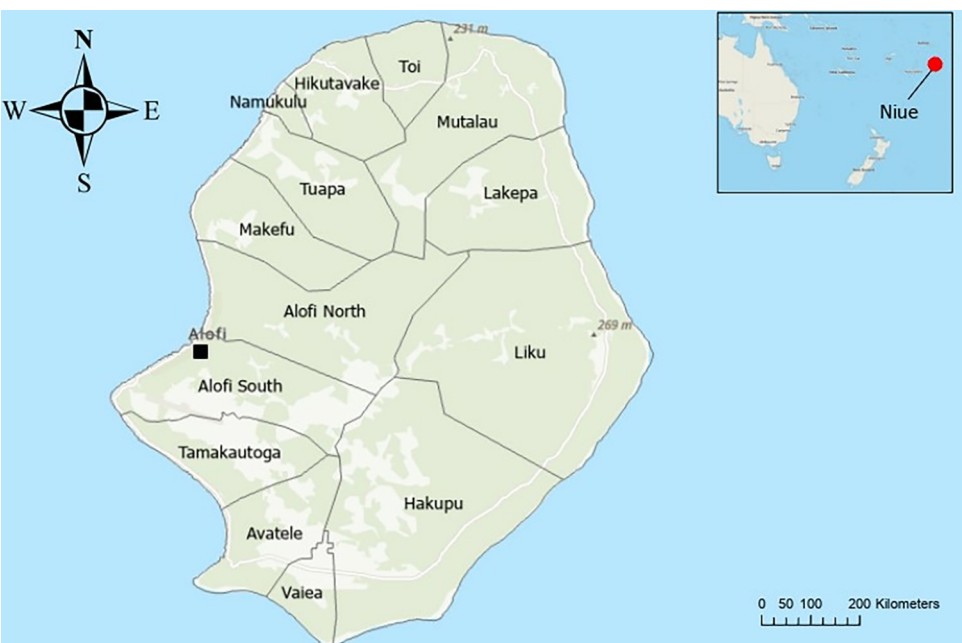

**Fig 1. Map of Niue showing administrative regions, the capital (Alofi) and Niue's location in the Pacific Ocean (insert).**

W) with a population of approximately 1,600 people [17]. The island covers approximately 261.46 km$^2$ and is divided into 14 administrative areas, with the capital, Alofi, divided into two districts: Alofi South and Alofi North (*Fig 1*) [17].

Basemap and map insert have been sourced from Toitū Te Whenua Land Information New Zealand and are licensed for reuse under CC-BY-4.0. The base map is available from https://data.linz.govt.nz/layer/52181-niue-island-polygons-topo-150k/. The image was developed using ESRI ArcGIS Pro.

Carlingford and colleagues (2019) provided a comprehensive overview of LF, its surveillance, and MDA activities in Niue [8]. Below we provide a summary.

The first Mf survey was conducted in Niue in 1954 with an estimated prevalence of 22.2% (166 of 748 adults surveyed) [18,19]. In 1956, MDA of diethylcarbamazine citrate (DEC) 50 mg administered monthly for at least 11 months was implemented [18] and a follow-up survey of people aged >6 years was conducted in December of the same year, finding Mf prevalence had reduced to 3.0% (83 of 2,791 people aged >6 years) [18]. Mf-positive individuals were treated with DEC 50 mg three times daily for two weeks with a week interval before a third week of treatment, continuing until individuals tested Mf-negative. In 1960 another Mf survey was conducted, revealing that Mf prevalence remained more or less unchanged at 3.2% (31 of 957 people surveyed) [20].

Twelve years later, in 1972, a third Mf survey revealed that Mf prevalence had risen to 16.4% (724 of 4,408 people surveyed) [21]. The survey was reported to have included 98.4% of the population. However, the specifics of the population subset used as the denominator for this calculation are unclear. A second round of MDA was implemented with DEC (6 mg/kg) administered once weekly for twelve weeks, followed by once monthly for twelve months [21].

A 1996 survey of 82% of the population revealed a Mf prevalence of 1.8% (26/1471) [22]; and, in 1997, a third round of MDA was undertaken with DEC (6 mg/kg) and ivermectin (200 μg/kg) to those ≥4 years of age; 87% of the population was reached [23].

In 1999, PacELF was launched and in September of that year, an LF antigen (Ag) prevalence survey was conducted on all residents aged ≥2 years; a population Ag prevalence of 3.1% (56 of 1,794 people surveyed; M = 42, F = 14) was reported [20]. A relatively high number (n = 13) of Ag-positive individuals were found in two locations (the villages of Tamakautonga and Hikutavake) that were distant to each other (i.e., not bordering each other). No children (<10 years old) tested Ag-positive; however, there were seven positives in the 10–19 years age group (n = 453). Ag prevalence was highest in those aged 20–29, 50–59, or 60+ years. Subsequently, MDA was administered annually from 2000 to 2004 (i.e., five rounds), with coverage rates reported as 94.2%, 99.1%, 82.2%, 77.5% and 85.2% of the population, respectively [20].

In September 2001, after the second round of MDA, a whole-population survey was undertaken that showed Ag prevalence had fallen to 1.3% (22/1630). Ag-positives were found in two villages where no cases had previously been detected, and no cases were found in Tamakautonga and Hikutavake, villages in which Ag-positives were found during the 1999 survey [20]. In 2002, a follow-up survey of Ag-positives identified in previous surveys (n = 20) revealed 12 tested Ag-positive. A further follow-up survey in 2003 of Ag-positives (n = 26) revealed 16 tested Ag-positive [20].

In August 2004, after the fifth round of MDA, a national 'stop MDA C-survey' was conducted in individuals older than 2 years which revealed an Ag prevalence of 0.2% (3/1285) [24]. Then, in 2009, a whole-population survey found an overall Ag prevalence of 0.5% (n = 1378) with no cases in six to seven-year-old children. The seven individuals who tested positive in 2009 were re-tested for Ag and treated the positives until testing Ag-negative.

Niue received WHO's validation of the elimination of LF as a public health problem in October 2016 [25]. In 2024, health officials reported that there have been no reported cases of lymphedema or filarial hydrocele in Niue for over two decades, and the last person with severe lymphedema died almost 35 years prior (G. Mokoia, personal communication, August 20,2024).

*Fig 2*, below, provides a graphical overview of LF surveillance and MDA in Niue.

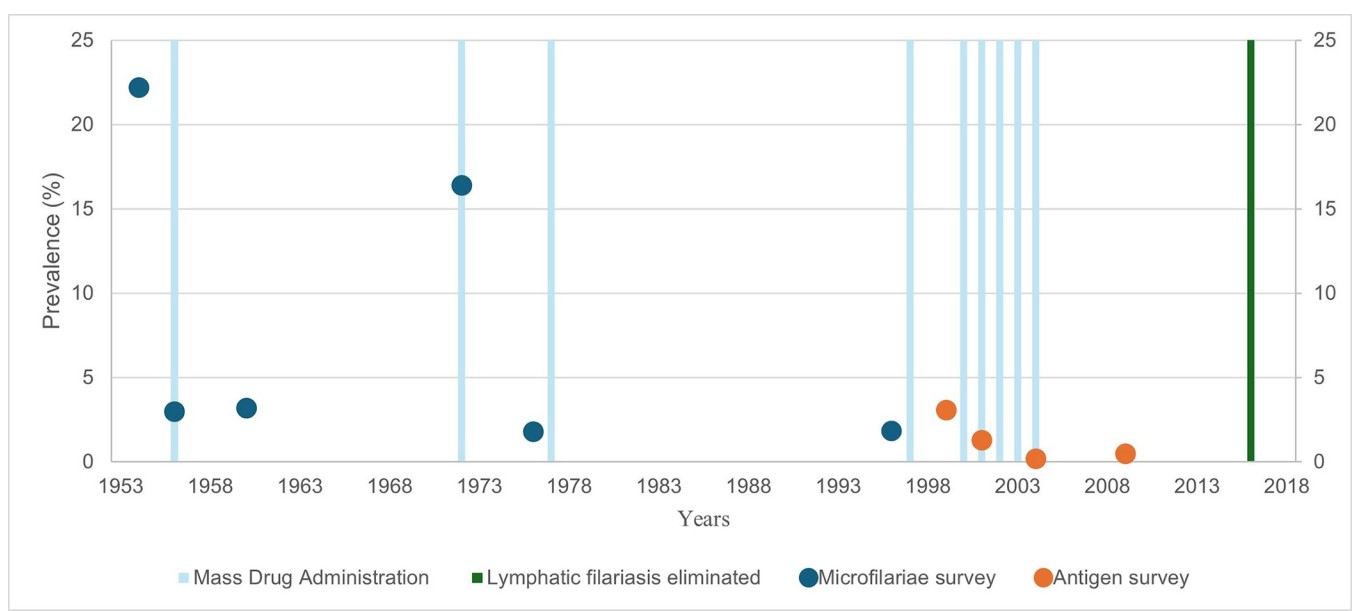

**Fig 2. Timeline of lymphatic filariasis surveillance results and mass drug administration in Niue, 1954 to 2016.**

## Study rationale

Countries that have received validation for eliminating LF as a public health problem are encouraged to conduct periodic PVS to ensure that LF transmission has not been re-established; however, to date there are no official guidelines as to how best to conduct PVS. The WHO *Neglected Tropical Diseases Roadmap 2030* [12] suggests transitioning from disease-specific approaches to more efficient and integrated methods for detecting, diagnosing, and treating neglected tropical diseases (NTDs) by 2030. Since its validation in 2016, Niue has not conducted PVS, and there are limited examples of integrated surveillance systems for eliminated or near-eliminated diseases in the Pacific Island Countries (PICs) or elsewhere.

This study seeks to determine if transmission is occurring eight years after validation of LF elimination as a public health problem. If Ag-positive participants are identified, it will enable rapid and targeted public health action to prevent the resurgence of LF in Niue.

Additionally, we will trial and investigate a novel approach to LF surveillance by integrating surveillance of LF and other eliminated, near eliminated and priority infectious diseases with the WHO STEPwise approach to non-communicable disease (NCD) risk factor surveillance (STEPS) scheduled in Niue. We hypothesise that this approach will yield several benefits. First, integrating LF with another population-wide disease surveillance survey will likely prove cost-effective, as financial and human resources needed for a standalone disease-specific (or infectious disease-specific) population-based survey will be shared across programmes. Second, by integrating NCD and infectious disease surveillance activities, we expect that public intrusion and imposition (in terms of time required, parallel consent processes, and multiple specimen collections) will be reduced. Third, collection and analysis of a broader set of demographic, biological and environmental risk data may be used to determine disease risk to aid both LF and NCD control. Fourth, an expected benefit of integrating LF testing with the STEPS survey is removing the need for collecting blood samples and demographic information in two separate surveys. This should help combat 'survey fatigue' and increase acceptability among communities. Demographic information collected during the STEPS survey will be relevant for the LF survey and will be used to identify any associations with LF Ag-positivity and seroprevalence to other Ag. We recognise that challenges may arise, including difficulties in modifying established survey processes to enable integration, reaching a consensus on the most suitable procedures and workflows, and potential conflicts when the prioritisation of field activities benefits one programme over another. We plan to identify and address these challenges through early and open communication and the development of joint implementation plans.

## Aim

The aim of this study is to investigate an integrated approach for PVS of LF and provide an evidence base to inform future PVS strategies in Niue and the broader region.

Additionally, we will:

- Estimate the prevalence of LF in Niue and, if cases are identified, describe the demographic characteristics and geographical distribution of infected individuals.

- Estimate the prevalence of other priority NTDs (LF, yaws and trachoma), vector-borne diseases (VBDs) (all four serotypes of dengue, chikungunya and Zika), and vaccine-preventable diseases (VPDs) (diphtheria, tetanus, measles and rubella) in Niue, and where relevant, produce evidence useful for monitoring progress towards national and regional disease elimination targets.

## Methods

### Sample size and study site selection

This study will be implemented as part of a national population-based STEPS survey, which will be conducted over three weeks in March 2025, to confirm the presence or absence of LF and other priority infectious diseases.

STEPS surveys typically collect data from individuals aged 18 to 69 years. The Niue DoH have extended this to include all people aged 15 to 69 years. Consequently, our study will align and collect data from all individuals in all 14 villages that are aged 15 to 69 years of age. According to the 2022 Niue Population Census [17], this equates to approximately 1,062 individuals (~63% of the total population) (Table 1).

Data collection will occur over three weeks in the mornings (before usual working hours) in each of the 14 villages in Niue (*Fig 1; Table 1*). In August 2024, the National health authority conducted pre-survey community engagement to explain the survey's purpose and approach and provide an opportunity for community members to ask questions. The schedule for field-based data collection has been communicated to community members through village talks, print media, and radio broadcasts.

### Eligibility criteria

All Niue residents (regardless of citizenship) aged 15 to 69 are eligible to participate. The survey will include migrants to Niue, as previous transmission was found in villages where migrant communities from neighbouring LF-endemic countries congregate [8]. Migrant and mobile populations are also considered a high-risk group for LF transmission [26,27]. Tourists and non-residents will not be included in the survey.

### Ethics

The study has been approved by the Research Committee of the Niue Department of Health and ratified by The University of Queensland's Human Research Ethics Committee (Project: 2024/HE001375).

**Table 1. Target survey sample size, by village.**

| Village | Total population | Target sample size (in the age class 15–69 years) |
|---|---:|---:|
| Alofi South | 423 | 267 |
| Alofi North | 187 | 118 |
| Makefu | 73 | 46 |
| Tuapa | 103 | 65 |
| Namukulu | 9 | 6 |
| Hikutavake | 39 | 25 |
| Toi | 32 | 20 |
| Mutalau | 77 | 49 |
| Lakepa | 95 | 60 |
| Liku | 74 | 47 |
| Hakupu | 180 | 114 |
| Vaiea | 81 | 51 |
| Avatele | 128 | 81 |
| Tamakautoga | 180 | 114 |
| **Total** | **1681** | **1062** |

Source: [17].

## Survey implementation

**Field logistics.** In each village, a 'survey station' will be set up in a central location. Parallel participant workflows will be established at the survey station to allow participants to be processed simultaneously through the STEPS and LF surveillance activities. Participants will be invited, one by one, to approach the survey station, where they will be greeted by a staff member. Once written consent is obtained, data collection will begin. This will involve a questionnaire component to collect demographic variables (including questions relevant to the STEPS and LF surveillance components), measurements of height, weight, and blood pressure, collection of urine specimens (for the STEPS survey), and the collection of blood (for both the STEPS and LF surveillance components).

*Fig 3* presents a diagram showing at what points LF PVS will integrate with the STEPS survey's workflow.

**Informed consent.** At the survey station, a survey team member will explain the purpose and process of data collection. A standard participant information sheet will be provided and written informed consent (using a template form) will be sought. The consent form will be signed by the participant and witnessed by a community member or Niue Department of Health staff member. For participants under 18 years of age, an assent process and parental or guardian consent will be collected. The consent form will have separate sections for the STEPS survey and the LF survey components to allow participants to opt in or out of either component.

**Questionnaire.** The standard WHO STEPS survey tool (www.who.int/publications/m/item/standard-steps-instrument) will be adapted to include one additional LF-related question: The additional question will be, "To which countries have you travelled in the last 5 years?" For this survey, a person will be considered a member of a household if they identify the house as their primary residence or slept at the house the previous night. Data, including

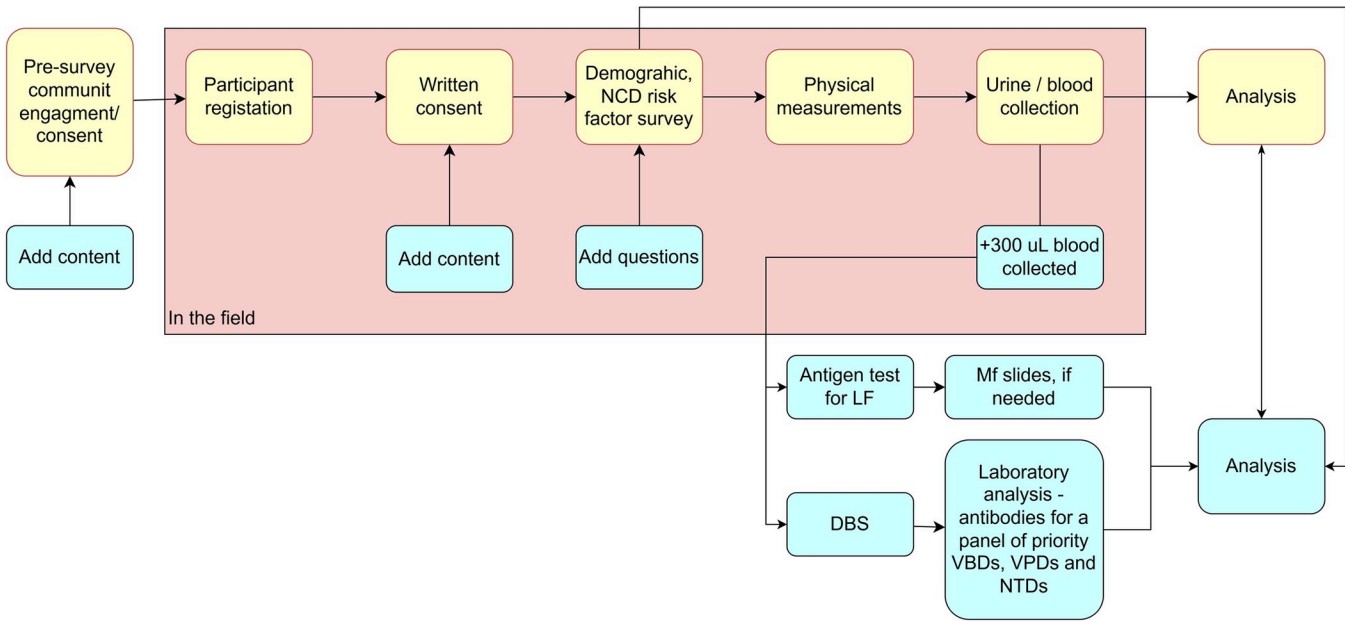

**Fig 3. Points at which post-validation surveillance for lymphatic filariasis will be integrated with the WHO STEPwise approach to NCD risk factor surveillance in Niue.** [Note that activities in yellow represent components of the STEPS survey; activities in the pink box represent activities undertaken in the field as part of data collection; and activities in blue indicate points where post-validation surveillance for lymphatic filariasis will be integrated with the STEPS survey's workflow].

house geolocations, will be entered directly into a purpose-built electronic data form on internet-connected tablets and automatically uploaded to a secure cloud storage facility.

**Body measurement and urine collection.** The STEPS survey team will conduct body measurements and urine collections as per their established protocols, which are not discussed in this paper.

**Blood collection.** For each enrolled participant, at least 300μL of blood will be collected by finger prick *(Fig 4Aa)* into heparin-coated BD Microtainer® Blood Collection Tubes *(Fig 4B)*. The tubes for the LF survey will be labelled and stored in a cooled box for analysis at the Niue Foou Hospital later the same day or the next day. If not processed immediately, blood samples will be kept in a fridge. Simultaneously, participants' blood glucose levels will be measured using a field glucometer.

**Laboratory testing.** Alere™ Filariasis Test Strips (FTS) (Abbott, Scarborough, ME) will be used to detect LF antigen (*Fig 4C*). For any Ag-positive samples, Mf slides will be prepared using previously described methods [28] (*Fig 4E*). Dried blood spots (DBS) will be prepared for all participants (irrespective of LF antigen positivity) for Multiplex Bead Assays (MBA) to detect anti-filarial antibodies using previously described methods [29,30] (*Fig 4D*). The same DBS will be used to test for antibodies against a panel of other NTDs, VPDs, and VBDs.

Laboratory data will be entered into a purpose-built electronic database using EpiCollect5.

**Participant, sample and data linkage.** To enable linkage of demographic and location data, questionnaire data, and FTS/DBS results collected in EpiCollect5, each participant will be issued a unique identifying number. These numbers and a corresponding quick response (QR) code will be printed on a strip of seven small stickers (*Fig 4D and 4E*). One sticker will be attached to each participant's consent form, questionnaire, blood collection tube, urine specimen containers, FTS, slides, and DBS.

**Data analysis plan.** The primary outcome measure for signalling the presence of LF in Niue will be a positive FTS. Crude Ag and Ab prevalence with 95% confidence intervals (CI) will be estimated using binomial exact methods. Seroprevalence of LF Ab will be estimated by

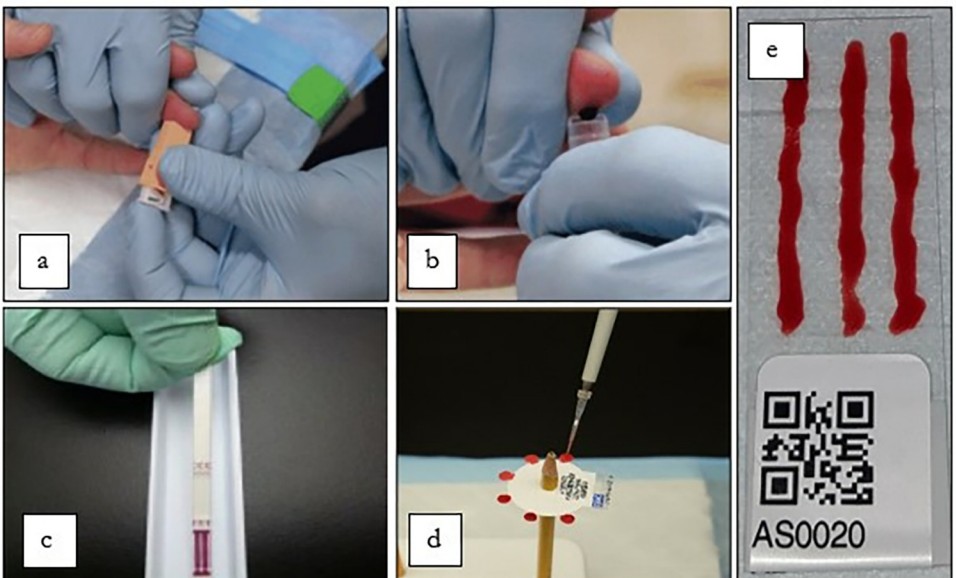

**Fig 4.** (a) Finger prick capillary puncture; (b) The collection of 300 μL of blood into a heparin-coated tube; (c) Filariasis test strip; (d) Dried-blood spots; (e) Microfilariae slides.

measuring IgG responses using MBA with Ag-specific cut-off values, determined by Median Fluorescence Intensity (MFI-bg), to establish seropositivity. Ag and Ab prevalence estimates will be adjusted for survey design and sex distribution in school-based surveys, and for survey design, sex, and age distribution in the community-based survey, based on Niue's census data [17].

Differences in demographic characteristics between Ag/Ab positive individuals will be described using mean ± standard deviation (SD), median [interquartile range], or number (per cent), and tested using Student's t-test or Mann–Whitney U test for continuous data and Pearson's chi-squared test of independence or Fisher's exact test for categorical data. Logistic regression will be used to assess any associations between demographic variables and Ag/Ab positivity. Variables with $p < 0.2$ on univariate analyses will be tested using multivariable logistic regression. Variables will be assessed using a variation inflation factor $< 5$ to check for potential collinearity, and final models will be selected using backward elimination, wherein variables are sequentially removed from the multivariable models to achieve the most parsimonious models, retaining variables with $p < 0.05$.

If possible, the sensitivity of Ag versus Ab to detect LF transmission in the post-validation period will be determined as the percentage of individuals with a positive FTS test among those testing Ab positive using MBA. The weights for agreement will be categorised as follows: $k < 0.00$ (no agreement), $k$ 0.00–0.20 (poor agreement), $k$ 0.21–0.40 (fair agreement), $k$ 0.41–0.60 (moderate agreement), $k$ 0.61–0.80 (substantial agreement), and $k$ 0.81–1.00 (almost perfect agreement). Lastly, seroprevalence estimates and mean MFI-bg values will be compared between communities with a history of LF transmission for significant differences.

## Discussion

Post-validation surveillance for LF should be established once national validation is achieved; however, initiating PVS as early as possible after the second transmission assessment survey is beneficial. While further evidence is required to determine the optimal duration of PVS following validation, due to the lifespan of adult worms, the general sentiment is to maintain surveillance for at least 10 years.

This study represents a valuable opportunity to assess the effectiveness of integrating PVS approaches for LF within a pre-established, pre-funded and periodic community-based survey (i.e., a STEPS survey).Given the high burden of NCD in the Pacific Islands [31], such surveys are likely to remain a regional priority for years to come. The WHO recommends STEPS surveys be conducted every 3 to 5 years [32]. We anticipate that the integrated approach outlined in this paper will be feasible and cost-efficient, providing a practical solution for the ongoing implementation of LF-PVS and surveillance for other eliminated, near-eliminated, and rare diseases in settings where resources are constrained and logistical challenges to population-wide surveillance are significant [33]. We expect the approach outlined here to generate interest as serve as a guide for use across the Pacific and–more broadly–the world.

The results of this survey will provide a comprehensive assessment of the status of LF and other priority NTD, VPD, and VBDs in Niue. This will provide insights relevant to national disease elimination, eradication, and surveillance programs. The findings of this study can help shape Niue's future LF-PVS and additional disease surveillance strategies. For example, the information generated may influence the Department of Health's decisions on whether future LF-PVS surveys are needed or if alternative methods, such as opportunistic testing of blood samples from donors, antenatal clinics or as part of routine blood tests should be explored. The seroprevalence information generated will also inform other disease response strategies; for example, understanding population dengue seropositivity by age group, location,

and serotype may inform future national dengue vaccination policy. Further, collecting information on the seroprevalence of VPDs will enable the identification of waning immunity and/or gaps in the immunisation landscape that can be addressed.

With regards to LF PVS, an Ag positive result will indicate a person is infected with a live or recently deceased adult worm and indicate a potential source of transmission. A Mf positive result will confirm that the person is carrying breeding adult worms and is a transmission source. We recognise that antibody assay analysis for LF is still in its early stages, and results are challenging to interpret. However, we plan to include LF antibody testing in the MBA panel, as previous studies (including those produced by out research team) [29,30,34,35] have shown that it can offer insights into epidemiology and help characterise pathogen transmission dynamics. Additionally, since parasite antigens are known to trigger an IgG response that is detectable over an extended period, the serological status of younger children (who should not have been infected with LF in their lifetime) could signal ongoing transmission [36]. The results of this survey will allow us to understand the status of LF in Niue. The information generated will be used to develop the next phase of activities and appropriate strategic responses. If no Ag-positive individuals are identified, we can be highly confident that LF has been eliminated as a public health problem in Niue; however, if there is low Ag prevalence or LF Ab are identified in individuals aged ≤20 years of age, our results will inform and direct an ongoing PVS strategy. If a high prevalence of Ag-positive or any Mf-positive individuals is identified, more intensive surveillance and targeted MDA will be considered.

While requiring adaptation to meet the local context, the protocol presented here provides a model others may apply. The results of our study (once implemented) will provide evidence for the feasibility of the approach and guidance for operationalisation. These insights will be valuable for those seeking to operationalise PVS in resource-constrained settings and meet the ambition to develop more efficient and integrated methods for surveillance and monitoring of NTDs set out in the WHO *Neglected Tropical Diseases Roadmap 2030*.

## Author Contributions

**Conceptualization:** Adam T. Craig, Harriet Lawford, Grizelda Mokoia, Minerva Ikimau, Colleen L. Lau.

**Funding acquisition:** Adam T. Craig, Harriet Lawford, Colleen L. Lau.

**Investigation:** Adam T. Craig.

**Methodology:** Adam T. Craig, Harriet Lawford, Minerva Ikimau, Peter Fetaui, Tonia Marqardt, Colleen L. Lau.

**Project administration:** Adam T. Craig, Harriet Lawford.

**Resources:** Adam T. Craig.

**Supervision:** Grizelda Mokoia, Minerva Ikimau, Colleen L. Lau.

**Visualization:** Harriet Lawford.

**Writing – original draft:** Adam T. Craig.

**Writing – review & editing:** Adam T. Craig, Harriet Lawford, Grizelda Mokoia, Minerva Ikimau, Peter Fetaui, Tonia Marqardt, Colleen L. Lau.

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
