## [Decision Letter · Decision Letter 0]

7 Oct 2024

PONE-D-24-36104Integrating post-validation surveillance of lymphatic filariasis with a non-communicable disease program in Niue, a study protocolPLOS ONE

Dear Dr. Craig,

Thank you for submitting your manuscript to PLOS ONE. After careful consideration, we feel that it has merit but does not fully meet PLOS ONE’s publication criteria as it currently stands. Therefore, we invite you to submit a revised version of the manuscript that addresses the points raised during the review process.

**ACADEMIC EDITOR: **I have invited four experts to review the protocol. Of them, three reviewers agreed to our request. All the reviewers considered this an important study in the context of LF post validation surveillance, but also raised a number of constructive comments. I would request the authors to concentrate and address the issues related to study methodology (hypotheses, sampling, etc) and Discussion.

We look forward to receiving your revised manuscript.

Kind regards,

Swaminathan Subramanian, Ph.D.

Academic Editor

PLOS ONE

Journal requirements: When submitting your revision, we need you to address these additional requirements. 1. Please ensure that your manuscript meets PLOS ONE's style requirements, including those for file naming. The PLOS ONE style templates can be found at https://journals.plos.org/plosone/s/file?id=wjVg/PLOSOne_formatting_sample_main_body.pdf and https://journals.plos.org/plosone/s/file?id=ba62/PLOSOne_formatting_sample_title_authors_affiliations.pdf 2. Please amend your list of authors on the manuscript to ensure that each author is linked to an affiliation. Authors’ affiliations should reflect the institution where the work was done (if authors moved subsequently, you can also list the new affiliation stating “current affiliation:….” as necessary). 3. Please include a caption for figure 3a 4. Please include a caption for table 2. 5. Please include a complete copy of PLOS’ questionnaire on inclusivity in global research in your revised manuscript. Our policy for research in this area aims to improve transparency in the reporting of research performed outside of researchers’ own country or community. The policy applies to researchers who have travelled to a different country to conduct research, research with Indigenous populations or their lands, and research on cultural artefacts. The questionnaire can also be requested at the journal’s discretion for any other submissions, even if these conditions are not met.  Please find more information on the policy and a link to download a blank copy of the questionnaire here: https://journals.plos.org/plosone/s/best-practices-in-research-reporting. Please upload a completed version of your questionnaire as Supporting Information when you resubmit your manuscript. 6. Thank you for stating the following financial disclosure:  [This work is funded by the Global Institute for Disease Elimination (FADE2024/03), the (Australian) National Centre for Immunisation Research and Surveillance (NCIRS) and Operational Research and Decision Support for Infectious Diseases Program at the Centre for Clinical Research, the University of Queensland (UQ). Staff from UQ led the study design, the decision to publish and the reparation of the manuscript.].  Please state what role the funders took in the study.  If the funders had no role, please state: ""The funders had no role in study design, data collection and analysis, decision to publish, or preparation of the manuscript."" If this statement is not correct you must amend it as needed. Please include this amended Role of Funder statement in your cover letter; we will change the online submission form on your behalf.

Reviewers' comments:

Reviewer's Responses to Questions

**Comments to the Author**

1. Does the manuscript provide a valid rationale for the proposed study, with clearly identified and justified research questions?

Reviewer #1: Yes

Reviewer #2: Yes

Reviewer #3: Yes

2. Is the protocol technically sound and planned in a manner that will lead to a meaningful outcome and allow testing the stated hypotheses?

Reviewer #1: Yes

Reviewer #2: Yes

Reviewer #3: Yes

3. Is the methodology feasible and described in sufficient detail to allow the work to be replicable?

Reviewer #1: Yes

Reviewer #2: Yes

Reviewer #3: Yes

4. Have the authors described where all data underlying the findings will be made available when the study is complete?

Reviewer #1: Yes

Reviewer #2: Yes

Reviewer #3: Yes

5. Is the manuscript presented in an intelligible fashion and written in standard English?

Reviewer #1: Yes

Reviewer #2: Yes

Reviewer #3: Yes

6. Review Comments to the Author

You may also provide optional suggestions and comments to authors that they might find helpful in planning their study.

Reviewer #1: This is a very clearly written study protocol. The proposed research is appropriate for the Niue context (which is highly unique) and should lead to information that benefits the national LF elimination effort - hopefully confirming that elimination status is maintained.

I would request that the authors address the following minor issues:

Study hypotheses do not appear to be unique. Particularly hypotheses 1 & 3. I suggest removing hypothesis 3, it feels like a subset of hypothesis 1.

Line 129 has a pair of errant words

Line 150: were should be "where"

Line 155: please state the age group sampled in the 'C survey'

Line 164: missing a space in the date

Line 260: I suggest you use "the house" and not "a house"; I would imagine most people slept in a house (unless you really do mean to distinguish between people sleeping in doors vs. outdoors)

Can you add a little more commentary in the discussion around the applicability of integrating with STEPS surveys? Are they always census surveys in the Pacific? To what other countries would you anticipate this methodology would extend?

Several times you mention priority diseases. Please share this list of diseases for Niue to give the reader an idea of what will be assessed, and possibly inspire readers with the types of diseases that can be assessed in an integrated manner.

Reviewer #2: The resurgence of LF following validation of elimination can happen, although it may take time. The authors for this study describe a protocol aimed at LF PVS through integration with other activities. Addressing the following comments will strengthen the paper.

Line 76. Ae. Cooki. cooki should be with a small "c".

Line 113: correct the spelling of ESRI ArcGIS

Line 125: Change remined to remained.

Lines 105/128/133/139/149: Check the population number. In the abstract and in line 105, a population of 1600 is reported. In line 128, 4,408 people were tested in 1972. Has the population declined? What accounts for that?

Line 129: Remove "nor" after whether.

Line 196: Correct the spelling of "Fourth"

- Why has the sampling population been restricted to those aged 15 - 69 years? A clarification for this will be useful.

- A description of the STEPS survey and how the LF PVS fits in will be useful.

- Are there other opportunities for integration that could be explored beyond the STEP survey

- Portions of the text described in the "Field Logistics" section should be moved to the "Informed Consent" section. There is some repetitions e.g. consenting procedures.

- Knowing that molecular xenomonioting results in higher infection prevalence in mosquitoes compared to humans, is there any consideration for this in the PVS surveys?

- It would be useful to provide copies of the tools used (e.g. questionnaires) as supplementary files to guide the reproducibility of the work by others.

- Some potential benefits of integration were listed. It might also be helpful for the authors to describe any challenges as part of the integration of the LF survey and STEP, even at the planning stage.

Reviewer #3: Comments on the paper entitled “Integrating post-validation surveillance of lymphatic filariasis with a non-communicable disease program in Niue, a study protocol”

PLOS ONE accepts submissions for publication of Study Protocols for any study type within the journal’s scope. This study Protocols describe detailed plans for conducting research, including the background, rationale, objectives, methodology, statistical plan, and organization of a research project. Publication of this protocol can be considered as the protocol plans to integrate and generate data on various indicators and interpretation of the results of LF-PVS.

Niue with a population of about 1600 people and antigenemia based mapping was done in 1999, repeat antigenemia survey in 2001, 2004 and 2009. The surveys covering population aged 2 years and above showed a decline trend in antigenemia prevalence from 3.1 to 0.5. MDA for LF elimination was carried out from 2000-2004. WHO validation was done in 2016. The country has a long history of LF control/elimination, since 1954 and monitoring and evaluation by conducting whole population survey. Therefore it is pertinent to trial with the new study protocol of PVS. The protocol is well written and presented. However, the following are the minor comments:

Comments

Abstract

1. Consider revising the title, specifying the disease for integration with LF. It can be “Integrating post-validation surveillance of lymphatic filariasis with a standard STEPs survey in Niue, a study protocol”

2. Line 24-29: Can be deleted. Start with LF situation in Niue.

3. Line 32-34: Rephrase the sentence as “However, no surveillance was conducted ever since validation to confirm infection rates have remained below WHO’s elimination threshold”.

4. Line 42: survey can be deleted

5. Line 47: Consider replacing “for ongoing LF transmission” with “for LF”. Prevalence of antigenemia not necessary indicate the ongoing transmission. It indicates both past and recent exposure.

6. Line 49: Justify antibody assay when we collect data on antigen and Mf. It may not reflect the ongoing transmission.

Background and study rationale

7. Line 64:what is the parasite species other than Wuchereria bancrofti prevalent in South Pacific countries? W.bancrofti is probably the only LF parasite prevalent in Niue [WHO. The PacELF way: towards the elimination of lymphatic filariasis from the Pacific, 1999-2005. Manila: WHO Western Pacific Region; 2006].

8. Line 73: this article “Zeldenryk, L.M., Gray, M., Speare, R., Gordon, S., Melrose, W., 2011. The Emerging Story of Disability Associated with Lymphatic Filariasis: A Critical Review. PLOS Neglected Tropical Diseases 5, e1366.. https://doi.org/10.1371/journal.pntd.0001366” can be cited as reference for LF is the second leading cause of chronic disability.

9. Line 76: Correct Ae. Cooki as Ae. cooki

10. Line 116: cite the reference of Carlingford [7]

11. Line 128-130: Suggest to rephrase the sentence “The survey was reported to have included 98.4% of a population, although nor whether this was for the entire population or a subset (e.g., those above a certain age) is unclear.”

12. Line 140-142: Give prevalence data, if available instead of number of cases. [A relatively high number (n=13) of Ag-positive individuals were found in two locations (the villages of Tamakautonga and Hikutavake) that were distant to each other (i.e., not bordering each othe].

13. Line 150: replace “were” by “where”

14. Line 156-157: add “in individuals older than 2 years” before [23]

15. Line 159-160: rephrase the sentence as “The seven individuals who tested positive in 2009 were re-tested for Ag and treated the positives until testing Ag-negative”.

Methods

16. Line 216: add “in all 14 villages” after diseases

17. Line 228: a column “total population” can be added before “Target sample size”. Add “in the age class 15-69 years after target sample size”

18. Line 231-232: Migrants can also be excluded

Study implementation

19. Line 241: the target sample size for whole population (excluding below 15 years of age) LF survey can be mentioned.

20. Line 245 & 258: Consider providing a supplement table for “A standard STEPs survey questionnaire”

21. Line 253: preferably witness from either the community or health staff and witness by a survey team member can be avoided

22. Line 254: Assent form for individuals 15-17 years may be required

23. Line 263: Whether GPS of the household will be recorded?

24. Line 270: add “for LF survey” after blood collection tubes

25. Line 285: will the DBS samples be sufficient to carry out a panel of tests of other NTDs, VPDs, and VBDs?

Discussion

Discussion brings out possible outcome of the study. However, discussion should focus on the timeline and how long these LF-PVS surveys if demonstrated feasible should be carried out. What changes are required when it is translated into a larger population? Discussion may also indicate the frequency and duration of these surveys and interpretation of the results. The implications of showing Ag/Ab/Mf positives and any possible thresholds of each indicator may be discussed.

7. PLOS authors have the option to publish the peer review history of their article (what does this mean?). If published, this will include your full peer review and any attached files.

Reviewer #1: No

Reviewer #2: **Yes: **Dziedzom K de Souza

Reviewer #3: **Yes: **Krishnamoorthy

---

## [Author Response · Author response to Decision Letter 0]

21 Oct 2024

No. Comment Response

Journal requirements

1 Please ensure that your manuscript meets PLOS ONE's style requirements, including those for file naming. 

The Requirements are met.

2 Please amend your list of authors on the manuscript to ensure that each author is linked to an affiliation. Authors’ affiliations should reflect the institution where the work was done (if authors moved subsequently, you can also list the new affiliation stating “current affiliation:….” as necessary).

Each author has an affiliation. 

3 Please include a caption for figure 3a 

Figure 4a (note that the number of figures has changed) includes a caption. It is “Fig 4(a). Finger prick capillary puncture”

4 Please include a caption for table 2. 

The manuscript does not include Table 2.

5 Please include a complete copy of PLOS’ questionnaire on inclusivity in global research in your revised manuscript. Our policy for research in this area aims to improve transparency in the reporting of research performed outside of researchers’ own country or community. 

The questionnaire has been uploaded to the manuscript management system.

6 Please state what role the funders took in the study. 

The following text has been added at around line 22. “The funders had no role in study design or decision to publish. They will not be involved in data collection or analysis. The funder had no role in the preparation of this manuscript”

7 Please review your reference list to ensure that it is complete and correct. 

The reference list has been reviewed.

Reviewer 1’s comments

1 I suggest removing hypothesis 3, it feels like a subset of hypothesis 1 

Hypothesis 3 has been removed, as suggested.

2 Line 129 has a pair of errant words 

This typo has been corrected. Thank you.

 Line 150 were should be "where" 

“were” has been replaced with “where” (around line 153).

3 Line 155 please state the age group sampled in the 'C survey' 

The age group has been added.

4 Line 164 missing a space in the date 

Space added.

5 Line 260 I suggest you use "the house" and not "a house"; I would imagine most people slept in a house (unless you really do mean to distinguish between people sleeping in doors vs. outdoors)

“a” changed to “the” at around line 295.

6 Can you add a little more commentary in the discussion around the applicability of integrating with STEPS surveys? Are they always census surveys in the Pacific? To what other countries would you anticipate this methodology would extend? Several times you mention priority diseases. Please share this list of diseases for Niue to give the reader an idea of what will be assessed, and possibly inspire readers with the types of diseases that can be assessed in an integrated manner. 

In response: additional content that:

• provides more details about the approach to integrating PVS for LF with STEPS, including a figure (Fig 3) showing the points in the STEPS workflow at which integration will occur (around line 272)

• highlights the WHO recommends STEP surveys be conducted every 3 to 5 years, and hence we expect that we expect the approach outlined to generate interest across the globe (around line 364-369)

• list the priority diseases that will be tested has been added around line 224.

Reviewer 2’s comments

1 Line 76. Ae. Cooki. cooki should be with a small "c".

 The typo has been corrected.

2 Line 113: correct the spelling of ESRI ArcGIS

 The typo has been corrected.

3 Line 125: Change remined to remained. 

The typo has been corrected.

4 Lines 105/128/133/139/149: Check the population number. In the abstract and in line 105, a population of 1600 is reported. In line 128, 4,408 people were tested in 1972. Has the population declined? What accounts for that? Yes, the population of Niue has declined dramatically. The 2022 census reports an on-island population of 1,681, down from 5,000 in the 1960s. The decrease is primarily attributed to Niue’s free association status with New Zealand. This status allows Niuean citizens to travel to, work, and study in New Zealand without restriction. There are an estimated 27,000 Niueans in Auckland.

To clarify the text around line 130, we have edited it to read “…the survey was reported to have included 98.4% of the population. However, the specifics of the population subset used as the denominator for this calculation are unclear.”

5 Line 129: Remove "nor" after whether. 

This typo has been corrected.

6 Line 196: Correct the spelling of "Fourth" 

This typo has been corrected.

7 Why has the sampling population been restricted to those aged 15 - 69 years? A clarification for this will be useful.

WHO recommends that the target population for a STEPS NCD risk factor survey be, at a minimum, all adults aged 18 to 69 years (WHO STEPS Manual). Niue DoH decided to include 15-18-year-olds in their STEPS survey. As our PVS work aims to integrate with the STEP survey, we have aligned our sample age. To explain this, we have edited the text at around line 234. It now reads, “STEPS surveys typically collect data from individuals aged 18 to 69 years. The Niue DoH have decided to extend this to include all people aged 15 to 69 years, and consequently, our study will align.”

8 A description of the STEPS survey and how the LF PVS fits in will be useful. 

On review, we feel that the LF PVS integration with STEPs is captured in the section titled ‘Survey implementation’ (around line 260). However, to address the reviewer’s suggestion, we have added the following text and associated figure at around line 272. “Figure 3 presents a diagram showing at what points LF PVS will integrate with the STEPS survey’s workflow.”

9 Are there other opportunities for integration that could be explored beyond the STEP survey

We expect so; however, discussing other integration options is outside this protocol paper's scope. We will be mindful to include a discussion about the pros and cons and the opportunities and challenges of PVS integration in our results manuscript.

10 Portions of the text described in the "Field Logistics" section should be moved to the "Informed Consent" section. There is some repetitions e.g. consenting procedures. 

Text in the ‘field logistics’ section provide an overview while details about how informed consent will be collected are presented in the ‘informed consent’ section. The brief introduction of consent in the field logistics section is needed to maximise clarity. As such, we have not made any major changes. Small edits have been made around lines 266 and 286.

11 Knowing that molecular xenomonioting results in higher infection prevalence in mosquitoes compared to humans, is there any consideration for this in the PVS surveys? 

Molecular xenomonitoring is not being considered in Niue at this stage. Also, it is not an obvious fit with the STEPS survey workflow and hence not a good fit for integration.

12 It would be useful to provide copies of the tools used (e.g. questionnaires) as supplementary files to guide the reproducibility of the work by others.

Ideally, we agree. However, the survey tools are still being drafted. We intend to include these in our results manuscript.

13 Some potential benefits of integration were listed. It might also be helpful for the authors to describe any challenges as part of the integration of the LF survey and STEP, even at the planning stage. 

In response, at around line 209 we have added “We recognise that challenges may arise, including difficulties in modifying established survey processes to enable integration, reaching a consensus on the most suitable procedures and workflows, and potential conflicts when the prioritisation of field activities benefits one programme over another. We plan to identify and address these challenges through early and open communication and the development of joint implementation plans.”

Reviewer 3’s comments

1 Abstract: Consider revising the title, specifying the disease for integration with LF. It can be “Integrating post-validation surveillance of lymphatic filariasis with a standard STEPs survey in Niue, a study protocol” 

We have amended the title. It now reads “Integrating post-validation surveillance of lymphatic filariasis with the WHO STEPwise approach to non-communicable disease risk factor surveillance in Niue, a study protocol” (line 1-4).

2 Line 24-29: Can be deleted. Start with LF situation in Niue. 

We have edited out much of the text the reviewer suggests be removed.

3 Line 32-34: Rephrase the sentence as “However, no surveillance was conducted ever since validation to confirm infection rates have remained below WHO’s elimination threshold”. 

The change has been made around line 34. The passage now reads, “ However, no surveillance has been conducted since validation to confirm infection rates have remained below WHO’s elimination threshold.”

4 Line 42: survey can be deleted 

The word “survey” has been deleted.

5 Line 47: Consider replacing “for ongoing LF transmission” with “for LF”. Prevalence of antigenemia not necessary indicate the ongoing transmission. It indicates both past and recent exposure.

The word “ongoing” has been deleted.

6 Line 49: Justify antibody assay when we collect data on antigen and Mf. It may not reflect the ongoing transmission. 

The following text has been added at around line 396. “We recognise that antibody assay analysis for LF is still in its early stages, and results are challenging to interpret. However, we plan to include LF antibody testing in the MBA panel, as previous studies [29, 30, 34, 35] have shown that it can offer insights into disease epidemiology and help characterise pathogen transmission dynamics. Additionally, since parasite antigens are known to trigger an IgG response that is detectable over an extended period, serological data from younger children (who should not have been infected with LF in their lifetime) could signal ongoing transmission [36].” 

7 Line 64: what is the parasite species other than Wuchereria bancrofti prevalent in South Pacific countries? W.bancrofti is probably the only LF parasite prevalent in Niue [WHO. The PacELF way: towards the elimination of lymphatic filariasis from the Pacific, 1999-2005. Manila: WHO Western Pacific Region; 2006].

8. Line 73: this article “Zeldenryk, L.M., Gray, M., Speare, R., Gordon, S., Melrose, W., 2011. The Emerging Story of Disability Associated with Lymphatic Filariasis: A Critical Review. PLOS Neglected Tropical Diseases 5, e1366.. https://doi.org/10.1371/journal.pntd.0001366” can be cited as reference for LF is the second leading cause of chronic disability. 

Good pick-up, thank you. We have removed the word “predominant” from the sentence, so it now reads, “W. bancrofti is the parasite responsible for LF”. The suggested reference has been added.

8 Line 76: Correct Ae. Cooki as Ae. cooki

This typo has been corrected.

9 Line 116: cite the reference of Carlingford [7]

”While Carlingford and colleagues are referred to in the sentence, we have added an in-text citation, as suggested.

10 Line 128-130: Suggest to rephrase the sentence “The survey was reported to have included 98.4% of a population, although nor whether this was for the entire population or a subset (e.g., those above a certain age) is unclear. The sentence has been edited. Thank you for the helpful comment.

11 Line 140-142: Give prevalence data, if available instead of number of cases. [A relatively high number (n=13) of Ag-positive individuals were found in two locations (the villages of Tamakautonga and Hikutavake) that were distant to each other (i.e., not bordering each othe]. 

Unfortunately, we do not have denominator data for these villages from that time. No change has been made.

12 Line 150: replace “were” by “where” 

The typo has been corrected.

13 Line 156-157: add “in individuals older than 2 years” before [23] 

This text has been added to the sentence.

14 Line 159-160: rephrase the sentence as “The seven individuals who tested positive in 2009 were re-tested for Ag and treated the positives until testing Ag-negative”. The suggested text has been used. Thank you.

 Method 

15 Line 216: add “in all 14 villages” after diseases 

This content has been added to the sentence.

16 Line 228: a column “total population” can be added before “Target sample size”. Add “in the age class 15-69 years after target sample size” 

The table has been edited to include the reviewer’s suggestions.

17 Line 231-232: Migrants can also be excluded The text around line 250 has been edited to read, “All Niue residents (regardless of citizenship) aged 15 to 69 are eligible to participate. The survey will include migrants to Niue, as previous hotspots of transmission were found in villages where migrant communities from neighbouring Pacific countries and territories congregate [7]. Migrant and mobile populations are also considered a high-risk group for LF transmission [25, 26]. Tourists and non-residents will not be included in the survey.”

18 Line 241: the target sample size for whole population (excluding below 15 years of age) LF survey can be mentioned. This detail is stated in the preceding section titled “Sample size and study site selection”. Further, we do not think it sits well under the heading “Field Logistics.” No changes have been made.

19 Line 245 & 258: Consider providing a supplement table for “A standard STEPs survey questionnaire” 

The questionnaire is in draft. We will include the finalised LF PVS integrated survey tool as a resource when we publish our results.

20 Line 253: preferably witness from either the community or health staff and witness by a survey team member can be avoided 

The text has been edited to read, “a community member or Niue Department of Health staff member.”

21 Line 254: Assent form for individuals 15-17 years may be required 

The text “an assent process” has been added to the sentence at around line 288.

22 Line 263: Whether GPS of the household will be recorded? 

The text “including house geolocations” has been added to the sentence at around line 296.

23 Line 270: add “for LF survey” after blood collection tubes 

The text “tubes for the LF survey” has been added to the sentence at around line 305.

24 Line 285: will the DBS samples be sufficient to carry out a panel of tests of other NTDs, VPDs, and VBDs? 

Yes, we have adopted a protocol that has been tried and tested elsewhere, including studies led by our team in other Pacific Islands.

25 Discussion brings out possible outcome of the study. However, discussion should focus on the timeline and how long these LF-PVS surveys if demonstrated feasible should be carried out. What changes are required when it is translated into a larger population? 

In response, two changes to the discussion section have been made:

• new content has been added near the start of the discussion section (around line 359) that reads, “Post-validation surveillance for LF should be established once national validation is achieved; however, initiating PVS as early as possible after the second transmission assessment survey is beneficial. While further evidence is required to determine the optimal duration of PVS following validation, due to the lifespan of adult worms, the general sentiment is to maintain surveillance for at least 10 years.” and

• new content has been edited at around line 411 to read, “While requiring adaptation to meet the local context, the protocol presented here provides a model others may apply. The results of our study (once implemented) will provide evidence for the feasibility of the approach and guidance for operationalisation. These insights will be valuable for those seeking to operationalise PVS in resource-constrained settings and meet the ambition to develop more efficient and integrated methods for surveillance and monitoring of NTDs set out in the WHO Neglected Tropical Diseases Roadmap 2030.”

26 Discussion may also indicate the frequency and duration of these surveys and interpretation of the results. The implications of showing Ag/Ab/Mf positives and any possible thresholds of each indicator may be discussed. 

The following text has been added at around line 393, “With regards to LF PVS, an Ag positive result 

---

## [Decision Letter · Decision Letter 1]

19 Nov 2024

PONE-D-24-36104R1Integrating post-validation surveillance of lymphatic filariasis with the WHO STEPwise approach to non-communicable disease risk factor surveillance in Niue, a study protocolPLOS ONE

Dear Dr. Craig,

Thank you for submitting your manuscript to PLOS ONE. After careful consideration, we feel that it has merit but does not fully meet PLOS ONE’s publication criteria as it currently stands. Therefore, we invite you to submit a revised version of the manuscript that addresses the points raised during the review process.

Both reviewers have agreed with most of the responses to their comments except the one of the comments of reviewer #2. As you will note from reviewer #2's comments, they are requesting that the questionnaires being used in the study are provided with the protocol at this point rather than with any subsequent research article for transparency purposes. From the IRB-approved protocol, there appears to be (i) a standard STEPs household questionnaire and (ii) the addition of one question “To which countries have you travelled in the last 5 years?”, which is being added to this. In view of this, please could you provide a copy of the STEPs household questionnaire or a link to the questionnaire, and add into the text details of the additional question, along the lines of  the 'Survey-based data collection' section of the IRB-approved protocol."

We look forward to receiving your revised manuscript.

Kind regards,

Swaminathan Subramanian, Ph.D.

Academic Editor

PLOS ONE

Reviewers' comments:

Reviewer's Responses to Questions

**Comments to the Author**

1. Does the manuscript provide a valid rationale for the proposed study, with clearly identified and justified research questions?

Reviewer #2: Yes

Reviewer #3: Yes

2. Is the protocol technically sound and planned in a manner that will lead to a meaningful outcome and allow testing the stated hypotheses?

Reviewer #2: Yes

Reviewer #3: Yes

3. Is the methodology feasible and described in sufficient detail to allow the work to be replicable?

Reviewer #2: No

Reviewer #3: Yes

4. Have the authors described where all data underlying the findings will be made available when the study is complete?

Reviewer #2: No

Reviewer #3: Yes

5. Is the manuscript presented in an intelligible fashion and written in standard English?

Reviewer #2: Yes

Reviewer #3: Yes

6. Review Comments to the Author

You may also provide optional suggestions and comments to authors that they might find helpful in planning their study.

Reviewer #2: I thank the authors for addressing my comments. However, on the comment of providing the survey tools, the authors' response is that these tools are now being drafted and will be made available as part of the results manuscripts.

If this is the case, then the submission should be placed on hold until the tools which are an integral part of the protocol are ready and submitted alongside. The principal reason for insisting on the tools is that there is no guarantee that the results manuscript will be submitted for publication or submitted to this journal. Also, the readers looking at this current submission need to be able to access all the tools without having to wait for a couple of years for the results manuscript nor look for it in another publication. Thus, there is no point submitting an incomplete study protocol for publication.

Reviewer #3: Reviewed the revised manuscript and the response to the comments (Reviewer 3). The revison is satisfactory. Except a comment, all the other comments have been accpeted by the authors and revised the manuscript accordingly. It is recommended that the manuscript in the present form may be accepted for publication.

7. PLOS authors have the option to publish the peer review history of their article (what does this mean?). If published, this will include your full peer review and any attached files.

Reviewer #2: No

Reviewer #3: **Yes: **Kaliannagounder Krishnamoorthy

---

## [Author Response · Author response to Decision Letter 1]

20 Nov 2024

Thank you for your message on 21 November asking that we provide written authorisation to use the map presented at Fig 1 in our manuscript.

We note that the map is provided by Toitū Te Whenua Land Information New Zealand and are licensed for reuse under CC-BY-4.0. This is clearly stated the website form which the map was obtained under the heading “license” on the right-hand side (see: https://data.linz.govt.nz/layer/52181-niue-island-polygons-topo-150k/).

To be 100% clear, we have edited the caption to the figure to read, “Basemap and map insert have been sourced from Toitū Te Whenua Land Information New Zealand and are licensed for reuse under CC-BY-4.0. The base map is available from https://data.linz.govt.nz/layer/52181-niue-island-polygons-topo-150k/. The image was developed using ESRI ArcGIS Pro.”

I trust this is sufficient to proceed with publication.

---

## [Editor Report · Decision Letter 2]

28 Nov 2024

Integrating post-validation surveillance of lymphatic filariasis with the WHO STEPwise approach to non-communicable disease risk factor surveillance in Niue, a study protocol

PONE-D-24-36104R2

Dear Dr. Craig,

We’re pleased to inform you that your manuscript has been judged scientifically suitable for publication and will be formally accepted for publication once it meets all outstanding technical requirements.

Kind regards,

Swaminathan Subramanian, Ph.D.

Academic Editor

PLOS ONE
---

## [Editor Report · Acceptance letter]

2 Dec 2024

PONE-D-24-36104R2 

PLOS ONE

Dear Dr. Craig, 

I'm pleased to inform you that your manuscript has been deemed suitable for publication in PLOS ONE. Congratulations! Your manuscript is now being handed over to our production team.

Kind regards, 

on behalf of

Dr. Swaminathan Subramanian 

Academic Editor

PLOS ONE